# HTA and Gastric Cancer: Evaluating Alternatives in Third- and Fourth-Line Patients

**DOI:** 10.3390/ijerph20032107

**Published:** 2023-01-24

**Authors:** Lucrezia Ferrario, Federica Asperti, Giuseppe Aprile, Jacopo Giuliani

**Affiliations:** 1Centre for Health Economics, Social and Health Care Management, LIUC Business School, HD LAB—Healthcare Datascience, LAB LIUC University Carlo Cattaneo, 21053 Castellanza, Italy; 2Department of Oncology, ULSS 8 Berica Hospital, 36100 Vicenza, Italy; 3Department of Oncology, ULSS 9 Scaligera, Mater Salutis Hospital, 37045 Legnago, Italy

**Keywords:** metastatic gastric cancer, health technology assessment, FTD/TPI, economic assessment, Italy

## Abstract

Metastatic gastric cancer (mGC) represents an economic and societal burden worldwide. The present study has two aims. Firstly, it evaluates the benefits and the added value of the introduction of trifluridine/tipiracil (FTD/TPI) in the Italian clinical practice, defining the comparative efficacy and safety profiles with respect to the other available treatment options (represented by the best supportive care (BSC) and FOLFIRI (5-FU, irinotecan, and leucovorin) regimens). Secondly, it assesses the potential economic and organizational advantages for hospitals and patients, focusing on third- and fourth-line treatments. For the achievement of the above objective, a health technology assessment study was conducted in 2021, assuming the NHS perspective within a 3-month time horizon. The literature reported a better efficacy of FTD/TPI with respect to both BSC and FOLFIRI regimens. From an economic perspective, despite the additional economic resources that would be required, the investment could positively impact the overall survival rate for the patients treated with the FTD/TPI strategy. However, the innovative molecule would lead to a decrease in hospital accesses devoted to chemotherapy infusion, ranging from a minimum of 34% to a maximum of 44%, strictly dependent on FTD/TPI penetration rate, with a consequent opportunity to take on a greater number of oncological patients requiring drug administration for the treatment of any other cancer diseases. According to experts’ opinions, lower perceptions of FTD/TPI emerged concerning equity aspects, whereas it would improve both individuals’ and caregivers’ quality of life. In conclusion, the results have demonstrated the strategic relevance related to the introduction of FTD/TPI regarding the coverage of an important unmet medical need of patients with metastatic gastric cancer who were refractory to at least two prior therapies, with important advantages for patients and hospitals, thus optimizing the clinical pathway of such frail patients.

## 1. Introduction

Gastric cancer (GC) is the fifth most common cancer and one of the most relevant causes of cancer death worldwide, following lung and liver cancer [1]. Despite a gradual decline over time in incidence in both high- and low-prevalence countries, GC remains a serious global and public health burden, with an important social and economic impact.

Even though surgical treatment represents the most prevalent curative treatment in stage I to III gastric cancer [2], more than 50% of patients are not eligible for surgery because of a late diagnosis [3]. For such patients affected by metastatic gastric cancer (mCG), chemotherapy represents the standard of care.

As per the European Society for Medical Oncology (ESMO) guidelines, taxane (docetaxel and paclitaxel), irinotecan, or ramucirumab, as single agents or in combination with paclitaxel, are the preferred second-line treatment (SLT) options for patients who have performance statuses 0–1 [4]. However, treatment options can become limited once resistance develops, and until recently, the only choices available after second-line treatment failure were to attempt another SLT option and/or continue with best supportive care (BSC).

The use of more active and less toxic treatments in the second line allows more patients to proceed with further therapies. Based on real-life data, around 18% of patients may receive third-line treatments and about 8.0% of patients receive fourth-line treatment [5].

Despite the management of mGC has improved over the past decade, with the availability of more evidence to support the efficacy of systemic treatment in refractory gastric cancer beyond second-line treatment, the use of third-line chemotherapy, however, raises controversial issues, including the need to identify specific patients that may benefit most from a third-line treatment to avoid a potential use of costly and/or toxic agents in the last 3 months of life [6].

In GC guidelines, the first- and second-line recommendations are supported by level I evidence based on efficacy and safety data from randomized trials or meta-analyses [4,7]. However, no global standard later-line therapy has been recommended and available treatments vary among different countries.

Until recently, there have been no good-quality data to support third-line treatment in metastatic GC: within the Italian real-life clinical practice, for example, docetaxel, irinotecan, or paclitaxel are administered as third- or fourth-line treatments for mCG patients, although their use is not supported by the literature on the topic.

Innovation plays a key role in the oncological setting. Recently, the introduction of trifluridine/tipiracil (FTD/TPI) in heavily pretreated mGC has improved rates of survival and it has become the treatment option to be offered as a third- or fourth-line therapy whose implementation is well supported by evidence in the literature [8]. The approval of such a drug, which is also used for the treatment of metastatic colorectal cancer, is based on the results of the randomized phase 3 trial known as TAGS [8], reporting significant improvement in overall survival (OS) and progression-free survival (PFS) in refractory GC and gastroesophageal junction cancer. Based on the above, the ESMO guidelines on GC were updated (eUpdate 4 November 2019) and third-line chemotherapy with TFD/TPI has been recommended for patients who are of performance status (PS) 0–1 [I, A].

Furthermore, the introduction of FTD/TPI in this setting may be related to an increase in costs, thus requiring an in-depth analysis of the added value of such innovative technology, considering not only OS or PFS but also considering the patients’ clinical pathway efficiency.

In light of the above considerations, the main objective of the present paper is the definition of the public health implications related to a consolidated use of FTD/TPI within the clinical practice, for the treatment of patients with mGC, in the third- and fourth-line of treatment (referring to microsatellite stability (MSS) or mismatch repair proficient (pMMR), more than 95% of mGC), thus identifying the therapeutic strategy with a higher added value, not only for the hospitals but also for patients, intercepting their needs and expectations. Specifically, the advantages and disadvantages of FTD/TPI will be compared with the treatment options traditionally used for the cure of this condition, represented by best supportive care and FOLFIRI (5-FU, irinotecan, and leucovorin), docetaxel, and irinotecan as the main therapeutical alternatives administered in the clinical practice, considering the specific Italian experience [5,9].

The achievement of the above-mentioned objective would try to answer the following policy question: “What are the potential advantages, with reference to the different domains typically investigated within health technology assessment activities, related to the routine administration of FTD/TPI, as a third- or fourth-line treatment, to be offered to mCG patients, in comparison with the most utilized treatment options, not only in terms of clinical efficacy and safety but also in terms of economic-organizational efficiency, taking into consideration the National Healthcare Service (hereafter NHS) point of view, thus leading to a patient’s overall clinical pathway optimization and an improvement in the perceived quality of life?”.

## 2. Methods

For the achievement of the above-mentioned objective, a health technology assessment (HTA) was conducted in the Italian setting, during the years 2021–2022, using the EUnetHTA Core Model [10], for the definition of the potential advantages and disadvantages related to the consolidated use of FTD/TPI within the Italian clinical practice, in the context of treatment of mGC.

The HTA was conducted assuming the Italian NHS point of view, and considering a time horizon of 3 months, thus being consistent with the duration of the treatment cycle of the therapeutic regimens under assessment for third- and fourth-line patients, as well as with the OS achieved within such therapeutic options [8].

Due to the multidimensional and multidisciplinary nature of HTA, several aspects were considered [10]:(i)general relevance of the disease;(ii)technical relevance of the technologies under investigation;(iii)safety;(iv)efficacy;(v)economic-financial impact;(vi)social and ethical impact;(vii)equity impact;(viii)legal impact;(ix)organizational impact.

For the deployment of the above HTA dimensions, three different data sources were used:
(1)scientific evidence derived from a narrative literature review, for the definition of comparative safety and efficacy indicators, as well as for the definition of the potentially eligible population to FTD/TPI treatment;(2)health economics tools for the economic evaluation of the patient’s clinical pathway, assuming the investigated drugs, and for budget impact analysis;(3)qualitative approaches, by means of the development of a specific online qualitative questionnaire through the LimeSurvey platform, which was sent to a mailing list composed of 25 healthcare professionals directly involved in the proper mGC patients care and treatment, and was filled in by 2 oncologists, 3 pharmacists, and 3 nurses.


The narrative literature review was conducted considering the following proposed PICO.

P = patient (s) or population = adult patients with metastatic gastric cancer, on third- or fourth-line treatment.I = “Intervention,” = trifluridine/tipiracil (FTD/TPI).C = “Comparators,” i.e., traditional treatment = traditional treatment used in the clinical practice, consisting of best supportive care (BSC), FOLFIRI (5-FU, irinotecan, and leucovorin), docetaxel, and irinotecan.O = outcomes of interest declined as “overall survival—OS”, “progression-free survival—PFS”, “12-month OS rate”, “12-month PFS rate”, “3-month OS rate”, “3-month PFS rate”, and “Adverse events occurrence rate”.

The literature evidence came from the systematic search of literature databases (Pubmed, Embase, and Cochrane Library search engines) up to February 2022. Search terms were: “Trifluridine/tipiracil”, “FTD/TPI”, “BSC”, “Best supportive care”, “FOLFIRI”, “docetaxel”, “irinotecan”, “Metastatic gastric Cancer”, “mGC”, “third-line therapy”, “fourth-line therapy”, “clinical effectiveness”, “overall survival”, “radiographic progression-free survival”, “adverse events”, and “quality of life”.

It should be noted here that the narrative literature review focused on papers describing the population affected by mGC assuming third- or fourth-line treatment options, produced in English Language and preferably a randomized-control trial. Evidence focusing on treating mGC patients with immunotherapies and monoclonal antibodies was excluded. Furthermore, both docetaxel and irinotecan regimes were accordingly excluded due to the lack of scientific evidence supporting the administration of such therapeutic options for mGC patients in third- and fourth-line therapy. Based on this consideration, the analysis focused on the comparison of FTD/TPI, BSC, and FOLFIRI active regimens (5-FU, irinotecan, and leucovorin).

Papers responding to the PICO and meeting both the search strategy and the inclusions criteria were consequently included and synthesized according to a PRISMA flow diagram [11], thus mapping out the number of records (in terms of papers) identified, included, and/or excluded, and the reasons for exclusion. Furthermore, the assessment of the scientific evidence included in the HTA was performed through the JADAD scale for randomized-control trials [12] and the NewCastle-Ottawa Scale [13] for cohort and observational studies, thus defining the potential risk of bias. These scales were useful to check the replicability and generalizability of the results obtained.

The included papers were then used to retrieve evidence-based information regarding the safety and efficacy profiles of FTD/TPI, BSC, and FOLFIRI.

For the deployment of the economic dimension, an activity-based costing analysis was conducted based on the standard clinical pathway derived from the clinical practice of three different Italian hospitals taking on mGC patients by means of a Delphi Method approach [14]. In particular, the patients’ clinical pathways, mapped and evaluated considering a 3-month time horizon, were divided into the following main phases.

The therapy choice phase, considering all the procedures and activities performed to choose the best therapy for the mCG patients, based on their clinical conditions.The drug administration phase, consisting of the costs related to the administration of the therapy based on the drug duration cycle, assuming a 3-month time period.The treatment monitoring phase, regarding the panel of examinations, visits, and diagnostic procedures aimed at monitoring the therapeutical success, allowing timely intervention of any adverse events or treatment-related complications that occur.The pathology monitoring phase, regarding the panel of examinations, visits, and diagnostic procedures aimed at monitoring gastric cancer progression.The management and resolution of drug-related adverse events phase.

For the above phases, the following direct healthcare costs were considered: (i) cost of outpatient services; (ii) cost of laboratory tests; (iii) cost of drugs; (iv) cost of any hospitalizations, especially with reference to the phase of management and resolution of drug-related adverse events.

Data were economically valorized taking into consideration the national outpatient and inpatient reimbursement tariffs valid for the year 2022. The cost of each pharmacological treatment is derived from the published Italian NHS price list, considering the national “ex-factory cost”. The recommended daily dose was determined according to the summary of product characteristics.

After the definition of the cost of an mCG patient treated with FTD/TPI, BSC, and FOLFIRI, a budget impact analysis (BIA) was implemented to define the economic sustainability related to higher use of FTD/TPI within the Italian clinical practice, based on the overall Italian eligible population, evaluating the healthcare expenditure evolution up to 3 months. The BIA was thus utilized to predict the economic and financial consequences of adopting a new technology. To design the BIA, a baseline scenario in which all the national patients were treated with BSC and FOLFIRI, as well as a lower utilization rate of FTD/TPI, was compared with two innovative scenarios, differing from a higher FTD/TPI penetration rate.

The market share used for the development of the analysis is detailed in Table 1 and derived from the clinical practice of the three different Italian hospitals involved in the analysis.

The last data source was the collection of healthcare professionals’ perceptions. Thus, a specific qualitative questionnaire was administered for the assessment of the ethical (in terms of accessibility to care), social, organizational, and legal domains, being consistent with the literature evidence on the qualitative HTA dimensions assessment [10,15]. The questionnaire was filled in by 8 experts involved in the treatment of mCG, who gave their comparative perceptions of the three therapies under assessment, according to an evaluation scale ranging from −3 (less performant therapy) to +3 (most performant therapy) [16]. The collection of healthcare professionals’ perceptions was useful to fill in gaps that are left unexposed by the literature evidence [17], implementing a typical social science qualitative approach.

The ethical aspects explored the following items: (i) access to care on the local level; (ii) access to care for people of a legally protected status; (iii) impact on the hospital waiting list; (iv) generation of health migrations; (v) existence of factor limiting the use of the therapy for specific groups of patients; (vi) iniquity. The social dimension required the professionals’ perceptions with regard to (i) ability of the therapy to protect patients’ autonomy; (ii) ability of the therapy to protect patients’ dignity; (iii) ability of the therapy to protect the patients’ religion; (iv) impact of the therapy on the social costs; (v) patients and citizens can have a good level of understanding of technology; (vi) impact of the therapy on patient satisfaction; (vii) impact of the therapy on patients’ quality of life; (viii) impact of the therapy on patients’ caregivers’ quality of life. The analysis of the legal domains required the healthcare professionals’ perception regarding the following items: (i) permission level of the drugs; (ii) need for inclusion of the drugs’ registry; (iii) fulfillment of the safety requirements; (iv) infringement of intellectual property rights; (v) need to regulate the acquisition of the drug; (vi) the legislation covers the regulation of the therapy for all categories of patients.

In conclusion, the organizational domain focused on: (i) additional staff; (ii) training course for all the healthcare professionals involved; (iii) education of patients and caregivers; (iv) hospital meetings required; (v) learning curve; (vi) compliance with hospital protocol; (vii) additional hospital spaces or furniture; (viii) impact of the drug on hospital waiting lists; (ix) impact of the drug on the organizational management of adverse events; (x) impact of the drug on the organizational management of the patient, in terms of follow-up monitoring activities; (xi) impact of the drug on internal processes; (xii) impact of the drug on the hospital connection process; (xiii) impact of the drug on hospital purchasing process; (xiv) impact of the therapy on the patient’s clinical pathway optimization; (xv) impact of the drug on the occupancy of chemotherapy chairs; (xvi) impact of the drug on the monitoring activities; and (xvii) impact of the drug on the treatment duration.

From a statistical perspective, the healthcare professionals’ perceptions were analyzed by means of a one-way ANOVA test, thus revealing the existence of statistically significant differences among FTD/TPI, BSC, and FOLFIRI treatment options, reporting only the significance between groups, without showing the results related to the post hoc analysis.

## 3. Results 

### 3.1. Results from the Literature Evidence

As detailed in Figure 1, out of the 44 papers screened, 5 met the inclusion criteria [8,9,18,19,20], which allowed the efficacy and safety dimensions to be presided over, with quality information, especially regarding the innovative drug represented by FTD/TPI. Most records were excluded because they focused on first- or second-line treatments, or because they considered patients with other oncological diseases.

As mentioned above, both docetaxel- and irinotecan-alone strategies were excluded given the lack of scientific evidence reporting on their administration within the mCG third- and fourth-line setting.

With reference to the only RCT included, namely the TAGS study and its two sub-analyses [8,18,19], study quality and risk of bias were assessed with the JADAD scale. In contrast, evidence regarding the FOLFIRI treatment regimen [9,20], being observational cohort studies without a control arm, was evaluated with the New Ottawa Scale. Whereas the use of FTD/TPI as a third- and fourth-line treatment option is grounded on high-quality evidence, no effective use of the FOLFIRI regimen emerged, demonstrated by a low-quality level of evidence.

From an efficacy point of view, Table 2 depicts the most relevant indicators, expressed by means of the OS and PFS. It should be noted here that information for the FOLFIRI regimen refers to a cohort of only 33 patients, which makes the data not robust or replicable.

In addition, Tabernero and colleagues (2021), in a sub-analysis of the TAGS study [19], distinguished the main endpoints related to the third and fourth lines of treatment when comparing BSC and FTD/TPI + BSC. Focusing on patients in the third line of treatment, a median OS for FTD/TPI of 6.8 months versus 3.2 months in the head of BSC (*p*-value = 0.0318) and a median PFS of 3.1 months for TFD/TPI and 1.9 months for BSC (*p*-value= 0.0004) emerged.

The same improvement was shown with reference to patients in fourth-line treatment: FTD/TPI lead to both a better median OS (5.2 months versus 3.7 months, *p*-value = 0.0192) and a better median PFS (1.9 months versus 1.8 months, *p*-value < 0.0001), with respect to BSC.

Table 3 depicts the rate of Grade 3 and Grade 4 adverse events, as well as their related economic evaluation, considering a 3-month period.

### 3.2. Results from the Economic Analysis

Table 4 depicts the economic evaluation of the clinical pathway of a patient administered with FTD/TPI, BSC, or FOLFIRI as third- or fourth-line treatment options.

Results from the BIA (Table 5) revealed that the NHS, assuming a 3-month time horizon, would require additional investment in the routinary utilization of FTD/TPI, ranging from 19% (Innovative Scenario #1) to 24% (Innovative Scenario #2), for the treatment of 1508 mCG patients potentially eligible for a third- or fourth-line treatment. The investment only related to the one-year time horizon could positively impact an additional overall survival rate for the patients treated with the FTD/TPI strategy.

### 3.3. Results from the Qualitative Assessment

Table 6 reports the healthcare professionals’ perceptions regarding the HTA’s qualitative dimensions.

As mentioned in the methods section, eight healthcare professionals were involved (two oncological clinicians, three nurses, and three pharmacists), with an average seniority of taking on patients with oncological disease of 6.7 years.

Based on a 7-item Likert scale, no differences emerged concerning the equity dimension (*p*-value > 0.05). However, healthcare professionals agreed on the limited access of FTD/TPI at the local level (1.00 versus BSC 1.88 versus FOLFIRI 1.88, *p*-value = 0.048), although its consolidated use may generate an improvement in waiting lists (1.75 versus BSC 0.00 versus FOLFIRI −1.38, *p*-value = 0.004) since FTD/TPI requires oral administration and not intravenous administration like FOLFIRI.

From a social and ethical point of view, thus assuming the mCG patients’ perspective, the professionals involved revealed a slight advantage of the innovative treatment, compared to the alternatives (1.58 versus BSC 1.25 versus FOLFIRI 1.36, *p*-value = 0.036). Specifically, FTD/TPI would improve both patients’ (1.75 versus BSC 1.00 versus FOLFIRI 1.25, *p*-value = 0.028) and caregivers’ (2.00 versus BSC 1.00 versus FOLFIRI 1.50, *p*-value = 0.011) quality of life, as well as patient satisfaction (1.50 versus BSC 1.00 versus FOLFIRI 1.25, *p*-value = 0.041).

The three therapeutical options under assessment could be considered superimposable in the measurement, evaluating the legal impact (*p*-value > 0.05), even if FTD/TPI represents the sole drug that could be administered for the treatment of mCG patients in third- or fourth-line treatment, as cited in the guidelines. This potential additional advantage does not emerge from the perceptions declared by the healthcare professionals involved in the qualitative part of the study.

Focusing on the organizational dimension, although FTD/TPI required organizational effort related to the training courses dedicated to medical and nursing staff, as well as internal meetings to communicate a potential organizational change, it emerged that FTD/TPI would be able to significantly improve internal processes (1.00 versus BSC 0.000 versus FOLFIRI −1.13, *p*-value = 0.005), as well as the overall patient clinical pathway (2.00 versus BSC 0.571 versus FOLFIRI 0.625, *p*-value = 0.009). In addition, FTD/TPI would optimize the management of chairs devoted to chemotherapy treatments (3.00 versus BSC 0.850 versus FOLFIRI −1.00, *p*-value = 0.001), with a consequent improvement in the hospital waiting lists (1.25 versus BSC 0.00 versus FOLFIRI −2.23, *p*-value = 0.032). This specific aspect is a potential dimension of interest, recently quantified and also introduced in the public tenders in the Italian context, demonstrating a process advantage related to a different drug route of administration.

According to this consideration, in the attempt to quantify the impact of the introduction of FTD/TPI on the accessibility to care, in terms of release in the occupancy of chemotherapy chairs, the innovative molecule would be able to lead to an advantage ranging from a minimum of 34% to a maximum of 44%, strictly dependent to FTD/TPI penetration rate, as reported in Table 1.

The possibility to free up organizational resources, based on a decrease of hospital accesses devoted to chemotherapy infusion, would give hospitals the opportunity to take on a greater number of oncological patients requiring drug administration for the treatment of any other cancer diseases. Thus, assuming a 3-month time horizon, on average, from 234 to 302 oncological patients could be additionally treated by Italian hospitals, depending on the minimum and maximum penetration rate.

## 4. Discussion

Recently, innovative therapies have significantly improved the prognosis of mGC patients. However, the third-line or fourth-line chemotherapy and its regimens for progressive mGC after the second-line chemotherapy is largely practice-related and not evidence-based: treatments vary considerably in terms of reagent selection and treatment plan. Clinical decisions about the mCG third- or fourth-line therapies are grounded on a consensus-based approach rather than an evidence-based approach due to the availability of limited high-quality scientific evidence demonstrating the achievement of successful patient outcomes. Despite the fact that within the clinical practice, docetaxel, irinotecan, or paclitaxel alone are administered for the treatment of such frail patients, no scientific evidence is available to support their use for third- or fourth-line treatments.

Thus, since an innovative drug, FTD/TPI, was recently approved as the prevalent therapy to be offered to mCG patients in third- and fourth-line treatments, this HTA activity tried to narrow the existing defining strengths and weaknesses of FTD/TPI with respect to the best supportive care or established alternatives in the clinical practice (FOLFIRI). Despite the fact that the analysis should have focused on a rigorous comparison of FTD/TPI versus BSC (the most utilized option to control signs and symptoms, before the TAGS trial [8]), the addition of FOLFIRI to the comparison is based on retrospective studies [9,20] and it represents an established alternative in daily clinical practice. However, the administration of FOLFIRI is not supported by high-quality evidence. Other possible therapeutic options in third-line for mGC (all derived from retrospective studies) could be irinotecan monotherapy [21], less used in daily clinical practice towards FOLFIRI, docetaxel [22], currently not used in relation to the combination of paclitaxel plus ramucirumab in second-line treatment (or ramucirumab monotherapy in the case of residual peripheral neurotoxicity derived from first-line treatment, which would preclude the use of taxanes (paclitaxel or docetaxel) in subsequent lines), or FLOT (5FU, leucovorin, oxaliplatin, and docetaxel) in a perioperative setting. However, there is scientific evidence that in pretreated mGC patients, salvage chemotherapy (SC) is tolerated and significantly improves OS when added to BSC [23].

Based on this consideration, the analysis reports the added value regarding the use of TFD/TPI versus BSC or FOLFIRI regimen, given the availability of scientific evidence in the third- or fourth-line treatment settings.

In this view, multidimensional analyses are becoming relevant, since besides efficacy and safety, efficiency, costs, and patients’ perspectives have emerged as important factors that impact the choice of drug treatment, particularly for anticancer drugs. Governments and healthcare providers have thus displayed an increasing interest in learning more about efficiency in resource allocation, thus representing a public health issue. This aspect could affect the procurement process, with a practical implication, requiring an additional effort to create evidence and information concerning organizational factors.

Results of the study reported that the main advantages of FTD/TPI are related to higher safety and efficacy, especially in terms of higher OS and PFS median months, with respect to the BSC and FOLFIRI. Furthermore, in this setting, both survival gain and quality of life improvement depend not only on the effectiveness of third-line chemotherapy but also on the side effects of the cytotoxic drugs [3]. The most common adverse events were mainly hematological (neutropenia, leucopenia, and anemia) and gastrointestinal (nausea, vomiting, and diarrhea), and were generally manageable with dosage modifications and/or supportive care [8,20]. Thus, FTD/TPI had a manageable tolerability profile in patients with heavily pretreated metastatic gastric cancer in the TAGS trial, thus being consistent with that seen previously in patients with metastatic colorectal cancer [24].

Additional benefits are found from both an organizational and a social impact, given the need for fewer hospital accesses for monitoring the progression of the disease, as well as for the relevant release of chairs devoted to chemotherapy treatment, as FTD/TPI is an oral therapy. This nature of the innovative drug may also represent a potential factor in increasing adherence to treatment, with a consequent positive impact on the overall mCG patients’ quality of life, thus leading to a two-fold advantage: organizational-logistic and psychological/clinical. The literature on the topic reports that a good level of quality of life emerged for mCG patients treated with FTD/TPI and there was a trend towards FTD/TPI reducing the risk of a deterioration in the quality of life with respect to the BSC [18].

The economic results reported in the present study, in terms of economic resource absorption, related to the management of an mCG patient assuming FTD/TPI are consistent with the literature available on the topic. For example, in Greece, the cost per patient for FTD/TPI and BSC was estimated to be EUR 6965 and EUR 1906, respectively [25], thus also reporting that the introduction of FTD/TPI as a third-line treatment option for patients with mGC was predicted to be associated with a limited budget increase for the Greek payer. The same results have been registered in Japan: from the perspective of the Japanese public healthcare payer, FTD/TPI is more cost-effective than nivolumab for patients with heavily pretreated metastatic gastric cancer [26].

Despite additional investment emerging in the routinary introduction of the innovative drug within the clinical practice, a relevant benefit emerged regarding equity, due to its capability to offer a valid treatment option to mCG patients refractory to a second-line treatment. Furthermore, the possibility to have an oral administration would make the home management of a patient in the terminal phase of his/her life more feasible. In fact, spending the last months at home until death is the wish of many patients and their families [27,28].

This consideration also acquires a strategic relevance in the light of the National Recovery and Resilience Plan (NRRP), within pillar #6 aimed at creating local networks, facilities, and telemedicine for local healthcare, as well as at promoting innovation, research, and digitalization of the national health service. As such, among its main goals, the Italian NRRP depicts the decrease of inequalities in access to care and advocates the possibility of home management, especially in the case of frail patients, overcoming the geographical heterogeneity in healthcare assistance and achieving a better efficacy of the assistance yielded by these services.

Furthermore, the results may be useful for the overall decision-making process at the institutional level, to take rational decisions, about the proper allocation of healthcare resources.

This study has some limitations. First, the study used an indirect comparison method when assessing the safety and efficacy profiles of the FOLFIRI regimen. This is because there were no head-to-head randomized controlled trials for comparison between FTD/TPI, BSC, and FOLFIRI together. Second, the economic evaluation of the mCG clinical pathway was based on assumptions, thus assessing a standard clinical pathway, rather than an observational study revealing the economic resources absorption per patient; however, the assumptions were based on the treatment policies adopted in real-world clinical practice. Third, the sample involved in the qualitative analysis may be considered small, thus paving the way for additional analysis involving a higher number of experts, thus making the results achieved more robust.

The current analysis was conducted from the payer perspective, and, as such, only direct costs were considered. Whereas a societal perspective may be worthwhile, indirect costs, such as patient time, caregiver costs, and lost productivity, to reflect opportunity loss for society, were not used in the current analysis, and thus are a relevant topic for further research. The analysis of the social costs would be further integrated with the assessment of patients’ and caregivers’ perceptions of their treatment experience, thus evaluating both their patients’ reported experience measures (PREMS) and their patients’ reported outcome measures (PROMS).

## 5. Conclusions

The present HTA analysis, reporting advantages and disadvantages of FTD/TPI, has demonstrated that this treatment option would cover an important unmet medical need for mCG patients who are refractory to at least two prior therapies, with important benefits for patients and hospitals, thus optimizing the clinical pathway of such patients.

## Figures and Tables

**Figure 1 ijerph-20-02107-f001:**
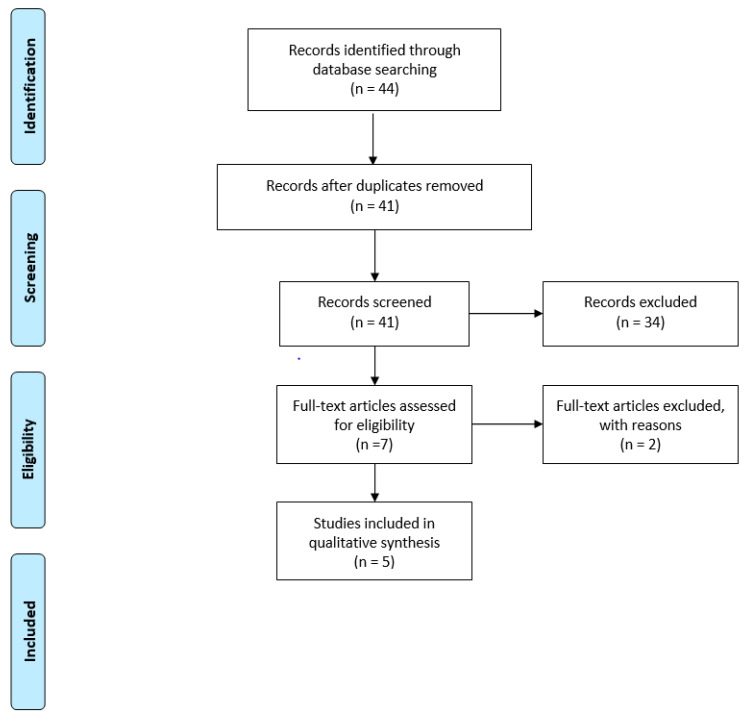
Prisma flow diagram.

**Table 1 ijerph-20-02107-t001:** Market shares used for the budget impact analysis.

	Baseline Scenario	Innovative Scenario #1	Innovative Scenario #2
FTD/TPI	10%	41%	50%
BSC	45%	29.5%	25%
FOLFIRI	45%	29.5%	25%

**Table 2 ijerph-20-02107-t002:** Efficacy indicators.

	FTD/TPI [8]	BSC [8]	FOLFIRI [9]
Median OS (months)	5.7	3.6	7.5
12-month OS (% of patients)	21%	13%	16%
Mortality rate considering a median follow-up equal to 10.7 months (% of patients)	72%	82%	n.a.
Median PFS (months)	2.0	1.8	3.3

**Table 3 ijerph-20-02107-t003:** Drugs’ related Grade 3 and Grade 4 adverse event occurrence rates.

	FTD/TPI [8]	BSC [8]	FOLFIRI [9,19]
Nausea	3%	3%	2.1% (based on the summary of product characteristics)
Anemia	20%	8%	6.30%
Decreased appetite	9%	6%	not applicable
Vomiting	4%	2%	3.03%
Diarrhea	3%	2%	9.09%
Fatigue	7%	6%	1.30%
Neutropenia	34%	0%	36.8%
Asthenia	5%	7%	3.03%
Thrombocytopenia	3%	0%	1.90%
Leucopenia	9%	0%	not applicable
Abdominal pain	4%	9%	not applicable
Constipation	2%	2%	1.90%
Back pain	1%	2%	not applicable
Increased blood alkaline phosphatase concentrations	3%	3%	not applicable
Dyspnea	2%	3%	not applicable
Dysphagia	3%	2%	not applicable
Ascites	4%	6%	not applicable
Hyponatremia	1%	4%	not applicable
Increased γ-glutamyl transferase concentrations	2%	3%	not applicable
Economic resources devoted to the management of drug-related adverse events	EUR 281.23	EUR 157.02	EUR 106.47

**Table 4 ijerph-20-02107-t004:** Economic evaluation of the mCG patient’s clinical pathway, considering a 3-month time horizon.

	FTD/TPI *	BSC *	FOLFIRI *
Therapy choice phase *	EUR 77.47	EUR 77.47	EUR 77.47
Drug administration phase *	EUR 5956.53	EUR 1443.29	EUR 5583.82
Treatment monitoring phase *	EUR 202.23	-	EUR 210.00
Pathology monitoring phase *	EUR 357.40	EUR 357.40	EUR 357.40
Management and resolution of drug-related adverse events phase *	EUR 281.23	EUR 157.02	EUR 106.47
Total Costs per treatment	EUR 6874.86	EUR 2035.17	EUR 6335.16

* Results based on real-life practices and expert opinions.

**Table 5 ijerph-20-02107-t005:** Budget impact analysis.

	**Baseline Scenario**	**Innovative Scenario #1**	**Difference (EUR)**	**Difference (%)**
FTD/TPI	EUR 1,036,729.25	EUR 4,250,589.92	EUR 3,213,860.67	310%
BSC	EUR 1,381,069.35	EUR 905,367.68	EUR −475,701.66	−34%
FOLFIRI	EUR 4,299,038.19	EUR 2,818,258.37	EUR −1,480,779,82	−34%
Total	EUR 6,716,836.78	EUR 7,974,215.97	EUR 1,257,379.19	19%
	**Baseline Scenario**	**Innovative Scenario #2**	**Difference (EUR)**	**Difference (%)**
FTD/TPI	EUR 1,036,729.25	EUR 5,183,646.25	EUR 4,146,917.00	400%
BSC	EUR 1,381,069.35	EUR 767,260.75	EUR −613,808.60	−44%
FOLFIRI	EUR 4,299,038.19	EUR 2,388,354.55	EUR −1,910,683.64	−44%
Total	EUR 6,716,836.78	EUR 8,339,261.55	EUR 1,622,424.76	24%

**Table 6 ijerph-20-02107-t006:** Analysis of the qualitative dimensions.

**Equity Impact**	**FOLFIRI**	**BSC**	**FTD/TPI**	***p*-Value**
Access to care on the local level	1.88	1.88	1.00	0.048
Access to care for a person of a legally protected status	2.00	2.00	2.00	Not applicable
Impact on the hospital waiting list	−1.38	0.00	1.75	0.004
Generation of health migrations	0.00	0.00	0.00	Not applicable
Existence of a factor limiting the use of the therapy for specific groups of patients	0.00	0.00	0.00	Not applicable
Iniquity	0.00	0.00	0.00	Not applicable
Average value for equity aspects	0.42	0.65	0.79	0.590
**Social and ethical impact**	**FOLFIRI**	**BSC**	**FTD/TPI**	***p*-value**
Ability of the therapy to protect patients’ autonomy	1.88	1.88	2.38	0.112
Ability of the therapy to protect patients’ dignity	2.38	2.48	2.38	0.459
Ability of the therapy to protect patients’ religion	1.63	1.63	1.63	Not applicable
Impact of the therapy on social costs	0.00	0.00	0.00	Not applicable
Patients and citizens can have a good level of understanding of technology	1.00	1.00	1.00	Not applicable
Impact of the therapy on patient satisfaction	1.25	1.00	1.75	0.041
Impact of the therapy on patients’ quality of life	1.50	1.00	2.00	0.028
Impact of the therapy on caregivers’ quality of life	1.25	1.00	1.50	0.011
Average value for social and ethical aspects	1.36	1.25	1.58	0.045
**Legal impact**	**FOLFIRI**	**BSC**	**FTD/TPI**	***p*-value**
Permission level of the drugs	0.00	0.00	0.00	Not applicable
Need for inclusion of the drugs’ registry	0.00	0.00	0.00	Not applicable
Fulfillment of the safety requirements	2.00	2.00	2.00	Not applicable
Infringement of intellectual property rights	0.00	0.00	0.00	Not applicable
Need to regulate the acquisition of the drug	1.13	1.00	−0.75	0.084
The legislation covers the regulation of the therapy for all categories of patients	1.00	1.00	1.00	Not applicable
Average value for legal aspects	0.69	0.67	0.38	0.379
**Organizational impact**	**FOLFIRI**	**BSC**	**FTD/TPI**	***p*-value**
Additional staff	0.00	0.00	0.00	Not applicable
Training course for all the healthcare professionals involved	−0.75	−0.75	−1.00	0.423
Education of patients’ and caregivers	−1.50	−1.38	−1.63	0.629
Hospital meetings required	−0.63	−0.71	−0.38	0.289
Learning curve	0.00	0.00	0.00	Not applicable
Compliance with hospital protocol	1.25	1.00	1.88	0.165
Additional hospital spaces or furniture	0.00	0.00	0.00	Not applicable
Impact of the drug on hospital waiting lists	−2.23	0.00	1.25	0.032
Impact of the drug on the organizational management of adverse events	−0.38	−1.50	−0.75	0.117
Impact of the drug on the organizational management of the patient, in terms of follow-up monitoring activities	0.38	0.43	0.38	0.319
Impact of the drug on internal processes	−1.13	0.00	1.00	0.005
Impact of the drug on the hospital connection process	0.38	0.00	0.25	0.543
Impact of the drug on the hospital purchasing process	0.38	0.00	0.50	0.678
Impact of the therapy on the patient’s clinical pathway optimization	0.63	0.57	2.00	0.009
Impact of the drug on the occupancy of chemotherapy chairs	−1.00	0.85	3.00	0.001
Impact of the drug on the monitoring activities	−1.00	0.25	0.75	0.074
Impact of the drug on the treatment duration	1.00	0.00	1.25	0.059
Average value for organizational aspects	−0.27	−0.07	0.44	0.031

Legend: all the items used for the enhancement of the qualitative HTA dimensions derived from both the EUnetHTA Core Model [10], IMPAQHTA framework [15], and the specific dimensions of interest for the evaluation proposed, collecting the experts’ perceptions.

## Data Availability

Data will be available from the corresponding author upon resoneable request.

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
