# Peer review of "HTA and Gastric Cancer: Evaluating Alternatives in Third- and Fourth-Line Patients"

_ijerph, 2023, doi:10.3390/ijerph20032107_

Round 1

Reviewer 1 Report

I think the paper is well structured and may be of importance for the oncologists in their every day decision making especially in coutries where the economical point of view is very critic. 

I think minor text editing are useful

About the paper “HTA and Gastric Cancer: evaluating alternatives in third- and fourth-line patients”

 I think it may be a useful paper/information for physicians treating patients with metastatic gastric cancer after second line chemotherapy treatment.

The main question addressed by the researches is the efficacy of the introduction, as a third and fourth line treatment in mGC, of trifluridine/tipiracil (FTD/TPI) in the Italian clinical day practice by comparing it with the actually treatment formulas ( Best Supportive Care -BSC- and FOLFIRI -5-FU, irinotecan and leucovorin- regimen).

The evaluation of the new treatment was done by comparing the OS rate towards Best Supportive Care -BSC- and FOLFIRI -5-FU  as much as actually there are no guide line recommendations for third and fourth line treatment in mGC patients.

Furthermore (second question of the paper) the authors have also evaluated the impact in terms of patients well-being and the economic impact (evaluating the indirect costs also) on the Italian NHS that the introduction of this new treatment strategy may have in a certain type of patient (patient with performance status 0-1).

I would suggest the authors, if possible, to integrate in the data tables also the p-value of the statistical studies that are mentioned at the results and discussion paragraph

The text is easy to read and understand. Although I am not an English mother language I think it needs some language corrections/editing.

Finally I think that authors correctly addresses the question posed and so the article can be published.

Author Response

Dear Reviewer,

We sincerely thank you for the positive and valuable comments about our paper.

Please find attached detailed replies to your requests.

Reviewer 2 Report

The manuscript is interesting and useful for health professionals concerned with oncology and gastric cancer treatment and public health decision-makers. The HTA model is appropriate to evaluate the benefits and limits of new treatments in oncology.

To go into details:

Introduction:

 I suggest that the authors give a short introduction of the different tasks that they will carry out in the study.

L26  Write the entire term before the acronym (mGC)

L47  Idem. For the European Society for Medical Oncology (ESMO)

L72 idem for TAGS

Methods:

 I suggest that the authors give a short explanation of the reasons why they have excluded immunotherapies and monoclonal antibodies from the narrative literature. This explanation will be helpful for readers that are not familiar with the treatment of metastatic gastric cancer.

I suggest that the authors give some more details to explain how was treated the innovative scenario 1 and 2 in the medico-economic analysis. Could they provide a reference to show that such a method has already been used in the literature?

The professionals were asked to fill a series of Likert scales on different HTA dimensions. The scales were analyzed with statistical methods.  I am not sure that I would call this work a qualitative analysis. Maybe calling this task a quantitative analysis would better fit with the work that was performed. Also including a description of the statistical analysis that was done on the results of the questionnaire could help improve the section.

Results:

Could it be possible to add more details regarding the literature review, and include the PRISMA flowchart?

 Could the authors give more details about the professionals who were interviewed? Socio-demographic data such as age and the number of years of experience would be useful.

Table 6. For some of the impacts in the table (first column) the sentences are written starting with a capital letter and others with a small letter, homogenize.

Discussion:

 “in terms reagent selection and treatment plan”: change to “In terms of reagent selection and treatment plan”

“in the light of the National Recovery and Resilience Plan”: I imagine that this is an Italian national work plan or guideline. I suggest that the authors give a short explanation of this national work plan.

I suggest that the authors compare their results with the current literature or mention that their work is innovative if no work has previously been done in this field in the literature.

In the limits of the work:

I suggest mentioning that the panel of professionals is small (statistical analysis was performed with a small number of people).

In perspectives

I suggest mentioning that it would be interesting to complete the social dimension with a qualitative analysis (including focus groups and interviews) based on the point of view of both patients and caregivers

Author Response

Dear Reviewer,

We thank you very much for reading our paper and for the positive comments received.

In the attachment, you could find all the replies to your suggestions.
